# “Health Outcomes of Grandparents Caring for Double Orphans in South Africa”: What Are the Determinants?

**DOI:** 10.3390/ijerph20247158

**Published:** 2023-12-08

**Authors:** Salmon Likoko, Monica Ewomazino Akokuwebe, Godswill Nwabuisi Osuafor, Erhabor Sunday Idemudia

**Affiliations:** 1Statistics South Africa, ISIbalo House, Pretoria 0002, South Africa; likokosalmon@gmail.com; 2Faculty of Humanities, North-West University, Mafikeng 2735, South Africa; erhabor.idemudia@nwu.ac.za; 3Department of Population Studies and Demography, North-West University, Mafikeng 2735, South Africa; gosuafor@gmail.com

**Keywords:** ageing, caregivers, demography, grandparents, gerontology, health, orphans

## Abstract

In the 21st century, grandparenthood is a significant phenomenon in the fields of demography, gerontology, and sociology. It is mainly explored in the context of ageing, as it is poised to become one of the most significant demographic phenomena and social issues in contemporary South Africa. Therefore, this study examined the determinants associated with grandparents who are parenting as caregivers and the health challenges they are exposed to as caregivers. The National Income Dynamics Study (NIDS) Wave 5 dataset was utilised, and a total of 302,476 grandparents aged 25 years and older, who were reported to be primary caregivers of double orphans, were included in the analysis. Both bivariate and multivariate binary logistic regressions were performed to determine the predictors of the determinants of grandparents parenting as caregivers and their health challenges in South Africa. Estimated odds ratios (ORs) with 95% confidence intervals (CIs) were used, and the threshold for statistical significance was established at ρ < 0.05. A majority of the male and female grandparent caregivers were aged 24–34 years, were Black Africans (69.8%), had secondary education (46.9%), reported health challenges (HC) (59.7%), with 26.4% reporting headaches in the last 30 days. Logistic regression revealed that grandparent caregivers aged 55–64 years were 8.9 times more likely to report health challenges compared to those aged 25–34 years. Non-Black African grandparent caregivers were found to be 0.61 times less likely to report health challenges, compared to Black African grandparent caregivers. Those with perceived poor health status were 3.3 times more likely to report health challenges, compared to those with excellent perceived health status. Therefore, there is an urgent need to redesign health interventions to address these health burdens among grandparent caregivers and to take cognisance of providing economic and social support for these vulnerable populations.

## 1. Introduction

In the 21st century, grandparenthood is a significant phenomenon in the fields of demography of ageing, social gerontology, and sociology, and it is mainly explored in the context of social aspects of ageing. It is poised to become one of the most significant demographic phenomena and social issues in contemporary South Africa [1,2]. Human immunodeficiency virus (HIV) and acquired immune deficiency syndrome (AIDS) have caused major social and economic devastation in South Africa, especially among children who have become orphaned by the disease. The premature death of parents due to HIV/AIDS leaves orphans without support, parental love, guidance, and resources, and this can be further followed by cycles of poverty, malnutrition, stigma, exploitation, and psychological trauma [3,4]. In many African societies and communities, such as in South Africa, the obligation for the care and welfare of orphans is placed on the closely connected members of the family, notably with the core expectation of these being the grandparents. The number of grandparents assuming the parental role of raising their grandchildren is becoming alarmingly high and has been showing a worldwide surge over the past 20 years [3,4]. Caregiving from grandparents ranges from primary care to co-care when living in an intergenerational home. In the absence of a parent, a group of grandparents, known as custodial grandparents, provide all of the child’s care in a household; such situations are prevalent in South Africa and commonly referred to as skip-generation households [2]. The extended families that once characterised the Black social structure have changed as a result of modernisation and urbanisation in both developed and developing nations, and family structures and functions have changed over time [5,6]. In a typical family, older members who had been a part of the extended family were replaced by a different form of family.

In addition, Fernandes et al. [7] reported the growing trend of grandparents parenting their grandchildren in the 1990s, which has also caught the attention of the press and policymakers in the United States of America. Previous studies have reported that 13.4% of the almost 7.1 million grandparent–grandchild households in the United States of America are custodial grand-families [8]. According to Meyer and Kandic [9], there has been an estimated 7% rise in custodial grandparenting in the United States since 2009. Since 1990, custodial grandparenting has increased in various low- and middle-income countries [4], including those in Africa [10] and Asia [11]. According to Buchanan and Rotkirch [12], and Nadorff and Patrick [13], approximately 1% of all children in the United Kingdom, and nearly 4.8 million children in the United States are raised by grandparents, while a study conducted by Hall et al. [14] reported that about 4 million children were being raised by grandparents in South Africa. The reason for such a situation is accounted for differently within each context and for different geographical locations. According to Buchanan and Rotkirch [15], the main reasons why children in the United Kingdom end up living with their grandparents were owing to an increase in parental desertion, death of parents, parents’ incarceration, rising drug abuse, and an increase in divorce rates [16]. In South Africa and other countries in Sub-Saharan Africa, the death of parents from HIV/AIDS has left many children in this situation to be raised by their grandparents [17]. Similarly, in Swaziland (formerly Eswatini), a majority of children in rural areas are forced to be under grandparental care as a result of poverty and HIV/AIDS [18]. 

South Africa has been experiencing changing family structures; the phenomenon has been evident even before the time of apartheid and has continued during the current democratic dispensation. However, one of the most noticeable changes in the family structure over the years is the transition from nuclear and extended families to a skip-generation family structure [1,2]. The transition in the family structure has been attributed to issues such as labour migration, non-marital childbearing, poverty, gender inequality, the death of parents, and neglect, among others [19,20]. HIV/AIDS has also played a significant role in changing the family structure, as orphaned children move in with family members, and grandparents in particular [21,22]. With these transitions, caregivers who are not the biological parents have taken on the responsibility of becoming informal caregivers to people living with disease or disability, and orphaned children [21,23]. Likewise, it has been noted that grandparents have been taking increasing responsibility for the primary care of orphans, in the absence of their biological parents in South Africa, resulting in what is known as grand-families [19]. From 1996 to 2011, the grandparent headship of households increased from 11.9% to 12.3%, showing an increase in the importance of grandparents’ contributions in South African households [19,24]. Moreover, in 2017, almost 2.7 million children were living with grandparent caregivers in the absence of their biological parents [1,25], with more female grandparents caring for orphans compared to male grandparents. Thus, grandmothers have become the new mothers, with transforming roles, signifying the existence and reality of grand-families in South Africa. Caregiving among grandparents is a moral and cultural obligation in African societies. The benefits of caregiving among grandparents have not gone unnoticed as they receive much satisfaction from parenting [26], and younger grandparents in the age cohort of lower than 40 years and who enter early into grandparenthood, have, to a certain extent, reported greater satisfaction as caregivers to their grandchildren [27]. 

However, the impact of grandparenting has yet to be significantly recognised and documented in South Africa. Caregiving from grandparents usually produces numerous benefits, such as having close-knit relationships with the children they are caring for. However, grandparents still face many difficulties, which include the role of caregiving, which is demanding, with insufficient or no formal caregiving training, and exposure to burdens in the form of physical, mental, social, and economic hardship [1,27]. Furthermore, grandparent caregivers often present with health challenges, such as poorer emotional well-being and declining psychological health, as a result of stressors arising from caring for their grandchildren [6,19]. A study conducted by Kidman and Thurman [21] among 726 caregivers of orphans in the Eastern Cape province revealed that 23% of caregivers reported having experienced chronic illness for three months or longer in the previous year. Also, another study conducted in Mankweng in Polokwane, among twelve grandparent caregivers of orphans, revealed that grandparent caregivers reported having hypertension, diabetes, and bodily aches owing to old age; one grandparent indicated that her poor health was a result of stress caused by her granddaughter [6,26]. The deterioration of the health of grandparent caregivers owing to stress is usually triggered by being unable to cope with the physical demands of raising small children and financial constraints. Moreover, in a qualitative study conducted in the Vhembe district in Limpopo province, grandparents were found to have reported experiencing anxiety, emotional stress, depression, bodily pain, hypertension, and high blood pressure when providing care to their grandchildren [1,6]. 

South Africa remains a complex mix of different races, cultural identities, languages, ethnic bonds, and social classes, as the country continues to have racial segregation. This racial segregation may perhaps have directly or indirectly created social concerns such as rape [28], pregnancy of children/adolescents [29], HIV and AIDS [30], tuberculosis (TB) [28], obesity [31], domestic violence [29], a high crime rate [29], unemployment [32], a high incidence of divorce [33], addiction to alcohol [30], and dependency on drugs or other substance use [34]. There is a dearth of studies on ageism conducted in South Africa, despite it being pervasive, and affecting people of all age cohorts, from childhood onwards, with serious and far-reaching consequences for individuals’ well-being, health, and human rights [35,36]. Ageism is typified by stereotypes (how one thinks about grandparents as caregivers), prejudice (how one feels about grandparents raising their grandchildren) and discrimination (how one acts towards grandparents giving care), which have a great impact on perceptions of other people based on their age. Because of the little attention ageism has attracted, issues associated with grandparenting and positive contributions by grandparents acting as caregivers to their grandchildren are not documented [37,38,39]. Adopting a better view of the core importance of grandparents playing the role of caregivers, despite having challenges as a result of caregiving, does not truly reflect the resilience of being a grandparent who takes up the challenges of parenting grandchildren in the absence of their biological parents.

The General Household Survey (GHS) showed that about 9% of children were paternal orphans, 3.1% of children (aged 0–17 years old) were maternal orphans, and 2.4% of children were double orphans [27,40]. Also, Statistics South Africa reported that the proportion of orphaned children in the KwaZulu-Natal province was 18.7%, one of the highest in South Africa [27]. Consequently, orphans have to rely on their ageing and often impoverished grandparents, who are not physically, emotionally, and financially ready for the new responsibility. This leaves grandparents with several challenges that they have to face, despite their incapacity to do so, which often has detrimental effects on their health outcomes [41,42]. This informs the underlying motivation for this study, as the range of health problems associated with grandparents carrying out caregiving has not been addressed. They are often the neglected portion of the population owing to stigmatisation if their children died as a result of AIDS, and from being in the ageing population with critical needs [18,27]. Few studies have been conducted in South Africa to examine health outcomes associated with grandparents acting as caregivers to orphaned children [19,36]. 

One neglected area of research is the determinants of health outcomes of grandparents caring for double orphans in South Africa. The purpose of this study is to examine the determinants associated with grandparents who are parenting as caregivers and the health challenges they are exposed to as caregivers to grandchildren who are double orphans. Findings from this study will be relevant to social gerontologists, demographers of ageing, and sociologists, and also to other health care practitioners (medical practitioners, community health workers, social workers, and public health experts); also, to policy-makers, as they will acquire knowledge through an in-depth understanding of this phenomenon from the study outcome, which will be of interest within the South African context.

## 2. Methods

### 2.1. Geographical and Sociodemographic Context of the Study

The study was conducted in South Africa, a country situated on the southern tip of Africa, bordered by Namibia, Botswana, Zimbabwe, Swaziland, and Lesotho. It has a land area of 1,221,037 square kilometres, stretching from latitude 22° S to 35° S and from longitude 17° E to 33° E [43]. One of the historical facts about South Africa is that the Dutch founded Cape Town in the country’s south in 1652, and Dutch farmers—known as Boers—started migrating there. Due to conflicts in Europe in 1806, the British took control of the Cape Town colony. The British merged the four local colonies to form the country of South Africa in 1910. The country has 3 capitals that operate at the executive (Pretoria), legislative (Cape Town) and judicial levels (Bloemfontein). It has a current population of 60.6 million people in 2022, with a life expectancy of 60 years for men and 67 years for women [43], with diverse cultures and population groups stratified as Black Africans, who are the majority (81.0%), followed by Coloureds (8.8%), Whites (7.7%) and Indian/Asian (2.6%) [43]. The country has nine provinces: Eastern Cape, Free State, Gauteng, KwaZulu-Natal, Limpopo, Mpumalanga, Northern Cape, Western Cape, and North West, and there are eleven official languages, namely isiZulu, isiXhosa, Afrikaans, English, Sepedi, siSwati, Sesotho, Setswana, Xitsonga, Tshivenda, and Ndebele. Recently, South African Sign Language became the twelfth official language. Notable demographics of South Africa include a fertility rate of 2.4 births per woman [44]. The economy has a GDP growth rate of 4.9% annually [43] and a Gross Domestic Product of USD 419 billion [45]. Although South Africa has one of the largest and most developed economies on the continent, the nation was previously governed by a white minority when the National Party came to power in 1948 and implemented its apartheid policy; this formalised the previously practised racial segregation. The discriminatory laws started to be overturned in the late 1980s, after decades of diplomatic isolation, military resistance, and large-scale protests. In 1994, the nation’s first nationwide elections for all races occurred. Despite efforts to address social injustices and promote reconciliation by the democratically elected administration, the economy continues to struggle. After the 1994 elections, the first post-apartheid population census was conducted in 1996, which included all people within the borders of South Africa [46]. The World Economic Forum warned in 2022 that South Africa is faced with a significant danger of state collapse amid records of extremely high unemployment rates, high crime rates, unaffordable government expenditure, poorly run institutions, and fraud [47]. In addition, South Africa still has the highest number of HIV infections globally and saw a dramatic increase in AIDS-related deaths, which peaked in around 2007. The estimated overall HIV prevalence rate is approximately 13.9% among the South African population, and people living with HIV (PLWHIV) is estimated at approximately 8.45 million in 2022, as well as an estimated HIV positivity rate of 19.6% among adults aged 15–49 years [43]. The HIV epidemic resulted in an upsurge in HIV morbidity and mortality among adults, and so childcare inevitably became part of grandparents’ activities [25,48]. In addition, nearly 2.7 million children in South Africa live in their grandparents’ households without their biological parents [25].

### 2.2. Study Design and Data Source

The 2017 wave 5 datasets from the National Income Dynamics Study (NIDS) were utilised in this study [49]. The Southern Africa Labour and Development Research Unit (SALDRU), established at the University of Cape Town’s School of Economics have conducted five waves of databases for the NIDS every two years, with the same household members, and they date from 2008 (wave 1), 2010–2011 (wave 2), 2012 (wave 3), 2014–2015 (wave 4), and 2017 (wave 5). NIDS is a nationally representative face-to-face longitudinal survey design comprising individuals residing in South Africa and their households, and it was initiated by the Department of Planning, Monitoring and Evaluation (DPME). This survey was conducted in order to understand the changing dynamics of poverty across the nine provinces in South Africa. Household living standards, household composition and structure, mortality, food and non-food spending and consumption, household durable goods, household net assets, agriculture, demographics, birth histories, children, parents and family support, labour market participation and economic activity, income and expenditure, grants, contributions given and received, education, health, emotional health, and household deprivation are among the research topics covered in the NIDS survey. The data for the NIDS survey were collected during the panel survey along with broad topics such as the household (household characteristics, household roster, mortality history, living standards, expenditure, consumption, adverse events, positive events, and agriculture); the adults (demographics, education, labour market participation, income, health, well-being, numeracy, and anthropometric data); and the children (education, health, family support, grants, anthropometric data, and numeracy). The NIDS started in 2008 with a nationally representative sample of over 28,000 individuals in 7300 households across the country. NIDS has Continuing Sample Members (CSMs) and Temporary Sample Members (TSMs), designed to follow individual members who are CSMs, wherever they may be in South Africa at the time of interview. Wave 5 includes proportions of respondents that were interviewed in earlier waves (71.9%), with 92% from wave 4, 87% from wave 3, 77% from wave 2, and 73% from wave 1 [50]. Within wave 5, the respondents such as the adults, children, household, and link files were merged, with the aim of having the characteristics of grandparent caregivers and those of the orphaned children and using the weight variable in the household file. However, the files were merged using a unique person identifier (PID); 400 enumerator areas were utilised, along with 7296 households selected to be part of the NIDS sample. Also, 300 fieldworkers were distributed around the nation’s nine provinces, to locate 28,226 individuals making up the selected households, and about 26,776 individuals were successfully interviewed throughout wave 1. In successive waves, the initial sample representatives are traced and re-examined; in the 2017 NIDS wave 5 datasets, 539,434 individuals were successfully interviewed. Figure A1 depicts the 5 waves of the National Income Dynamics Study (NIDS), illustrating the number of respondents in each wave and the selection process for the study population.

### 2.3. Study Population and Sample Size

The 2017 National Income Dynamics Study (NIDS) Wave 5 datasets were utilised, and a total of 302,476 grandparents aged 25 years (to be a grandparent at 25 involves 2 generations becoming pregnant at age 12 or lower) and older were reported to be primary caregivers in the datasets. 

However, the study population chosen was those grandparents who reported that they were caregivers to double orphans, and they were further stratified by the sex of the double orphans in the analysis. These are people who reported that they provided caregiving without being remunerated [51]. The number of grandparents providing care to male double orphans was 141,671 and female double orphans was 160,805, totalling 302,476 grandparent caregivers. The grandparents who reported to be primary caregivers were determined using a variable from the dataset “relationship code of the person responsible for the care of the child”. However, the variable had different relationship codes but only respondents who reported to be a grandparent or great-grandparent remained as the sample; other relationship codes were dropped. To ascertain that the children cared for were double orphans, only those who reported having lost both parents through death remained valid for this study. 

### 2.4. Variable Measurements

#### 2.4.1. Outcome Variable

The outcome variable of the study is health challenge, with a binary category outcome of Yes or No, and it was coded as Yes = 1 and No = 0. This was completed in order to carry out binary logistic regression analysis [52,53]. However, the health challenge was generated from a question that asked about some health conditions that people complain about at times. The question asked, “In the last 30 days, have you experienced […]?” with listed health conditions being fever, persistent cough, cough with blood, chest pain, body ache, headache, backache, joint pain/arthritis, diarrhoea, painful urination, swelling of ankles and severe weight loss. Those that reported Yes were coded as 1 and those who reported No were coded as 2. The study recoded all those coded as 2 to 0 (No) and those coded as 1 remained so (Yes). 

#### 2.4.2. Independent Variables

Independent variables (or factors) were selected for this study based on the objectives of this study and on review of existing studies [6,18], with consideration of the information available in the 2017 NIDS Wave 5 datasets. The independent factors were categorised as demographic, economic, health-related, and geographical type. First, the demographic variables were age (25–34 *, 35–44, 45–54, 55–64, and 65+), sex (male * and female), population group (Black African * and non-Black African), education (no education *, primary, secondary, and post-secondary). However, marital status was excluded from the regression analysis due to sample size reduction and multicollinearity of the predictor variable. Second, economic variables were assessed as regular salary (yes * and no), and pension (yes * and no). Third, health-related factors included in the analysis were depression in the past week (no * and yes), perceived health status (excellent *, good, poor), last health consultation (never *, in the last month, and last year or more) and medical aid (yes * and no). Fourth, geographical type variables were geographical area (rural * and urban) and province (Western Cape *, Eastern Cape, Northern Cape, Free State, KwaZulu-Natal, North West, Gauteng, Mpumalanga, and Limpopo). However, employment status, health index, geographical type, province, health insurance cover, and the individuals’ perception of health were explored as background characteristics and further used as determinants of health in the binary logistic regression model. All the variables are categorical. Note that the asterisk signs in the parentheses “*” indicated the reference category used in the binary logistic unadjusted and adjusted odds ratio in the analyses.

### 2.5. Inclusion and Exclusion Criteria

#### 2.5.1. Inclusion Criteria

The inclusion criteria involve the characteristics of the population that were included in this study. The inclusion criteria comprise males and females who are 25 to 65+ years of age, are grandparents, and have reported one or more forms of health conditions. 

#### 2.5.2. Exclusion Criteria

The exclusion criteria comprise the characteristics of the study population that do not meet the inclusion criteria, which may interfere with the outcome of the study. The exclusion criteria include male and female respondents who are less than 25 years of age, who are not grandparents, and did not report any form of health conditions. 

### 2.6. Statistical Analysis 

Datasets from the National Income Dynamics Study (NIDS) wave 5 [49] were adjusted for weighting to account for variations in sample probabilities, such as under- and over-sampling errors resulting from previous studies before this study analysis. Also, all analyses were carried out based on the outcome of interest and stratified by gender of the target population (caregivers of both male and female double orphans). Stata 14 statistical software version was employed to carry out data cleaning in order to detect and correct inaccurate, duplicate, or incomplete data within the wave 5 dataset. Analyses were completed in three phases: univariate, bivariate, and multivariate. First, the univariate analysis was performed to describe the characteristics of the outcome, and the independent variables (demographic, economic, health-related, and geographical type) associated with grandparent caregivers were presented in a table (Table 1). Graphs were drawn to show the percentage distribution of grandparents as caregivers to double orphans by sex, age, and population group (Figure A2, Figure A3 and Figure A4). Similarly, other graphs were drawn to show the prevalence of grandparents with health challenges as caregivers to double orphans by sex, age, and population group (Figure A5, Figure A6 and Figure A7). Similarly, a separate univariate analysis was conducted to show the proportion of health challenges reported by grandparent caregivers to double orphans (Table 2). Second, bivariate analysis, which employed the chi-square test, was performed to test the associations between grandparents caring for double orphans by sex and the associated factors (Table 3). Third, multivariate binary logistic regression was performed to evaluate the unadjusted and adjusted relationship between the outcome and explanatory factors, accounting for the effects of all other explanatory variables which are included in the regression models. Multicollinearity was checked using the “vif” command in the Stata software; the mean vif was 1.40 and is presented in tables (Table 4 and Table 5).

## 3. Results

### 3.1. Socio-Demographic Characteristics 

Table 1 above shows the demographic, economic, health-related, and geographical characteristics of grandparents as caregivers, stratified by caregiving to males (*n* = 141,671), to females (*n* = 160,805) and to both sexes (*N* = 302,476) of double orphans (Table 1). Demographics show the majority of grandparent caregivers of female double orphans were aged 25–34 years (26.6%), female (66.6%), and had secondary education (51.3%), while grandparent caregivers of male double orphans were mainly Black Africans (70.9%). With economic factors, grandparent caregivers for female double orphans reported they had a pension (81.7%), and grandparent caregivers for male double orphans reported no regular salary (65.4%) (Table 1). 

Similarly, by health-related factors, grandparent caregivers for female double orphans mentioned not having medical aid (81.2%), and grandparent caregivers for male double orphans reported having depression in the past week prior to the survey (42.9%), having poor perceived health status (22.1%), and had their last health consultation in the last month prior to the survey (65.5%). Finally, by geographical type, grandparent caregivers for females (41.0%), males (21.6%), and both sexes (30.7%) of double orphans were predominantly found in the Gauteng province (Table 1).

#### 3.1.1. Percentage Distribution of the Grandparents as Caregivers to Double Orphans by Sex, Age, and Population Group

Figure A2 shows the percentage distribution of grouped age of grandparents caring for double orphans. The findings revealed that grandparents aged 25–34 years were mostly caring for female double orphans, and grandparents aged 55–64 years were mainly found caring for male double orphans (See Appendix B). Similarly, Figure A3 illustrates the proportion of each sex of grandparents caring for double orphans. The findings revealed that female grandparents were found caring for double orphans irrespective of their sex (caring for males—60.8%, caring for females—66.6%, and caring for both sexes—63.9%) rather than male grandparents (See Appendix B). Figure A4 depicts the percentage distribution of the population group of grandparents caring for double orphans. Grandparents caring for double orphans were mostly Black Africans (caring for males—70.8%, caring for females—68.8%, and caring for both sexes—69.8%) (See Appendix B). 

#### 3.1.2. Prevalence of Grandparents as Caregivers to Double Orphans with Health Challenges by Sex, Age, and Population Group

Figure A5 shows the prevalence of grandparents with health challenges as caregivers to double orphans by sex in South Africa. A majority (38.5%) of female grandparents reported caring for both sexes of double orphans, while 21.4% of them cared for female double orphans, and 17.2% of them reported caring for male double orphans (See Appendix B). Figure A6 demonstrates the prevalence of grandparents caring for double orphans by age. Overall, 14.5% of grandparents aged 65+ years reported caring for both sexes of double orphans, while 7.3% of grandparents aged 45–54 years stated caring for female double orphans, and 7.6% of grandparents aged 55–64 years reported caring for male double orphans (See Appendix B). Figure A7 shows the prevalence of grandparents caring for double orphans by population group. This study’s findings showed that caring for male (9.6%), female (9.8%), and both sexes (19.4%) double orphans was lower among non-Black African grandparents (See Appendix B).

#### 3.1.3. Health Challenges Reported by Grandparents as Caregivers to Double Orphans

Table 2 above shows health challenges reported by grandparents as caregivers for double orphaned grandchildren prior to the survey in the last 30 days. The study’s findings revealed that grandparents as caregivers caring for double orphans mainly reported they experienced health conditions such as joint pain/arthritis (19.6%), backache (19.9%), body ache (20.1%), fever (26.4%) and headache (26.9%) (Table 2). Some others mentioned health conditions they were concerned with, including diarrhoea (6.9%), chest pain (7.5%), swelling of ankles (9.4%), and cough (9.6%) (Table 2). 

#### 3.1.4. Bivariate Analysis of Grandparents’ Caring for Double Orphans and Its Associated Factors by Sex

Table 3 presents the significant findings of the bivariate analysis involving grandparents caring for double orphans by sex, and its associated factors (demographic, economic, health-related, and geographical type) (Table 3). From the demographic factors, the findings revealed that grandparents aged 65+ years were found caring for males (95.4%) and both sexes (87.7%) of double orphans. Also, grandparents aged 55–64 years reported caring for males (79.0%), females (80.4%), and both sexes (79.6%) of double orphans. Age was found to be significantly associated at *p* < 0.05. Similarly, 72.8% of grandparents caring for female double orphans and 70.3% of them caring for both sexes of double orphans reported having no education. Also, 67.1% of them caring for female double orphans and 76.4% of them caring for both sexes of double orphans reported having primary education. Education was found to be significantly associated at *p* < 0.05. By economic factors, 73.9% and 65.6% of grandparents caring for males and both sexes of double orphans reported not having a regular salary. A regular salary was found to be significantly associated at *p* < 0.05 (Table 3). Likewise, 93.6% of grandparents caring for male double orphans reported having a pension, while 72.3% of grandparents caring for female double orphans stated having a pension, and 82.8% of grandparents caring for both sexes of double orphans reported having a pension. A pension was found to be significantly associated at *p* < 0.05 (Table 3). By health-related factors, grandparents caring for male, female and both sexes of double orphans reported a good (81.4%, 67.4%, and 74.7%) and a poor (83.3%, 88.6%, and 85.8%) perceived health status, respectively (Table 3). Perceived health status was found to be significantly associated at *p* < 0.05. Lastly, grandparents caring for males (79.0%) and both sexes (70.8%) of double orphans reported never having gone for a health consultation, while grandparents caring for males (69.4%), females (64.8%) and both sexes (67.0%) of double orphans had their last consultation in the last month. The last health consultation was found to be significantly associated at *p* < 0.05 (Table 3).

#### 3.1.5. Unadjusted Predictors of Health Challenges Experienced by Grandparent Caregivers

The significant predictors of health challenges experienced by grandparent caregivers in the unadjusted logistic regression analysis were—age (55–64 years and 65+ years), no regular salary, no pension, poor perceived health status, and health consultation (Table 4). According to the unadjusted binary regression model, the factors that significantly increased the likelihood of health challenges experienced as a result of being a caregiver to male orphans were—increased age 65+ years (unadjusted odds ratio (UOR) 8.70; *p* < 0.05), no regular salary (UOR 2.12; *p* < 0.05), poor perceived health status (UOR 5.87; *p* < 0.05) and health consultation in the last month (UOR 2.15; *p* < 0.05). For grandparents caring for female orphans, a significant likelihood of health challenges experienced was found among respondents with increased age 55–64 years (UOR 7.51; *p* < 0.05), poor perceived health status (UOR 5.79; *p* < 0.05), and health consultation in the last month (UOR 8.77; *p* < 0.05). A significant probability of health challenges experienced by grandparents caring for both sexes include factors such as no regular salary (UOR 1.90; *p* < 0.05), poor perceived health status (UOR 5.73; *p* < 0.05), and health consultation in the last month (UOR 15.74, *p* < 0.05) (Table 4).

#### 3.1.6. Adjusted Predictors of Health Challenges Experienced by Grandparent Caregivers 

The significant predictors of health challenges among grandparent caregivers to all double orphans were age, education, regular salary, pension, perceived health status, and health consultation (Table 5). The adjusted binary regression model has shown significant factors such as increasing age (55–64 years and 65+ years) among grandparent caregivers to all double orphans. For education, a significant likelihood of health challenges experienced was found among respondents caring for female orphans (AOR 13.94; *p* < 0.05) and both sexes of orphans (AOR 4.79; *p* < 0.05). Respondents with no regular salary caring for males (AOR 4.68; *p* < 0.05) and both sexes (AOR 2.01; *p* < 0.05) have higher odds of experiencing health challenges. Grandparents with no pension caring for both sexes (AOR 3.95; *p* < 0.05) of double orphans have increased odds of experiencing health challenges. Respondents with good perceived health status caring for males (AOR 6.15; *p* < 0.05) and both sexes (AOR 2.98; *p* < 0.05) have higher odds of experiencing health challenges. Also, respondents with poor perceived health status caring for males (AOR 4.28; *p* < 0.05), females (AOR 4.00; *p* < 0.05) and both sexes (AOR 3.30; *p* < 0.05) had increased odds of experiencing health challenges. Hence, regarding health consultation, grandparents caring for both sexes of double orphans were 12.87 times more likely to have experienced health challenges (AOR 12.87; *p* < 0.05) (Table 5).

## 4. Discussion

The results of the 2017 wave 5 of the NIDS are presented in this study, from nationally representative data, carried out to keep track of the well-being of South Africans [49]. This study indicated that grandparents in the age cohorts of 55–64 years and 65+ years experienced a higher prevalence of health challenges than those in the age groups of 25–34 years and 35–44 years. Further, grandparent caregivers of female double orphans reported the highest prevalence of health challenges, compared to grandparent caregivers of male, and both female and male double orphans. Also, the prevalence of health challenges remained highest among Black African grandparent caregivers of all double orphans. The observed prevalence of health issues among grandparents who are caring for their grandchildren after their parents pass away from HIV/AIDS is an indication that South Africa has not made much progress towards the SDG 1, SDG 2, and SDG 3 targets [48,54]. In a high-income country, family support is often passed down through the generations, especially from parents to children, and this significant kind of help includes looking out for the ages that follow. Grandparents continue to be an essential source of child care for many working parents, even though the number of children they manage has decreased as formal child care has increased.

Meanwhile, the proportion of grandparents who raise their grandchildren has grown over time [18,20]. Some grandparents step in to raise their grandchildren when the parents cannot do so owing to illness, drug addiction, or being in prison [38,54]. Also, other grandparents share custody of their grandchildren in response to their adult child’s financial need, separation and divorce, or employment commitments, as well as the death of one or both parents due to health conditions such as HIV/AIDS, tuberculosis, etc. [55,56,57,58]. Grandparents caring for grandchildren provide a critical provision and a fruitful platform for their grandchildren. The benefits of using grandparents to care for or raise grandchildren are both public and private, much like those of other forms of caregiving. Using grandparents to raise or care for grandchildren, particularly after the death of parents, preserves public resources and avoids discussions about public duty. However, as the importance of grandchild care has grown, concerns have surfaced that the benefits, as mentioned earlier, may jeopardise the well-being of grandparents [38,58], and the influence on grandparent’s health by caring for double orphaned grandchildren is a major focus of concern in this study. 

Therefore, this study found that cohorts of grandparents of increased age as caregivers to double orphans suffered many health challenges, as they are solely responsible for the well-being of their grandchildren [59,60]. We also found significant differences as their age increased, when looking at the health challenges experienced by these grandparents. This finding is consistent with another study conducted by Spinelli et al. [61], a study which found grandparents derived satisfaction as caregivers to their grandchildren despite experiencing other social problems. Grandparents play an important role in family life and it is culturally acceptable to have grandparents as caregivers across Sub-Saharan African nations such as Nigeria [62], Ethiopia [63], Malawi [55], and Mozambique [54]. Furthermore, our findings support the assumption that when parents are unable or unwilling to care for their children, grandparents are the first option. To reduce the effects of children growing up without parents, grandparents should be encouraged and supported to take on caregiving duties and parental roles for the grandchildren [61]. Also, findings from this study can be generalised to a bigger population as a result of the sample scope included in this study [59,64]. Additionally, in-depth research is required to identify the difficulties and issues that are being faced by grandparents as caregivers, especially in this era of both non-communicable and communicable diseases, such as tuberculosis and HIV/AIDS [19,42], considering the high prevalence of young parents of children who are out of work [61,62]. Thus, several studies have documented positive responses from studies that have worked with grandparents as caregivers to their grandchildren, despite the challenges they faced during the process of caregiving. As such, it would be very important to create and develop strategic strength-based interventions to tackle all the challenges plaguing grandparents as caregivers [54,59]. 

Attempts should be made to assist and allow grandparents to raise their grandchildren in cases when both parents have died, rather than trying to dissuade them from taking on the role of guardian and proxy parent [19,54]. To address some of the health challenges faced by grandparents, resources, such as in the social, financial, and health areas should be provided to reduce the pressure and fatigue related to the grandparents’ contribution to the parental role [19,42]. Strongly encouraging healthy intergenerational ties will reduce the abuse and desertion of elderly people such as grandparents [60,65]. Furthermore, our results showed that health conditions experienced by grandparents when providing caregiving to double orphans include joint pain/arthritis, backache, body ache, fever, and headache. Other health concerns such as chest pain, swelling of ankles, and a cough were mentioned by grandparents in this study. Caregiver burnout can occur in grandparents, producing a state of physical, emotional, and mental exhaustion. Stressed grandparent caregivers may experience fatigue, anxiety, and depression when providing parental care to grandchildren [65]. After all, being a grandparent serving as a caregiver is highly demanding, making it difficult for the carer to tend to their own needs first. Also, studies have shown that providing care can have a severe impact on one’s physical and mental health, leading to negative emotional effects, and poor treatment of the orphaned grandchildren they are caring for [66]. Also, other studies have mentioned that grandparents as primary caregivers stated depression, anxiety, changes in appetite (such as eating too much or too little), hypertension, cardiovascular disease, and chronic fatigue as health conditions they were suffering from as a result of attending to the needs of their orphaned grandchildren. These aforementioned health conditions may be caused or aggravated by the demands and necessities of caregiving to double orphans [67]. In addition, there is a critical need to conduct research that will look at an extensive review of health conditions and the health risks for harmful medical issues that may arise among grandparents providing care for double orphans in South Africa.

Furthermore, demographic (age, education), economic (regular salary, pension), and health-related factors (perceived health status, health consultation) in the unadjusted and adjusted models of the multivariate analysis of this study have been shown to influence the health conditions of these grandparents, and this assertion is in keeping with the findings of other studies [7,38]. In this context, this study found that among grandparents as caregivers, those aged 55+ years caring for male double orphans had greater odds of experiencing health conditions compared to those aged 25–34 years, and this result is supported by several studies [38,66]. This may be due to the fact that with the increasing age of older people, their bones tend to shrink in size and density, weakening them and making them more susceptible to fracture. Generally, in older people, their muscles tend to lose strength, endurance, and flexibility, which can affect their coordination, stability, and balance. Also, at the genetic level, ageing results from the impact of the accumulation of a wide variety of molecular and cellular damage over time [60,66]. Stress and exhaustion from caregiving can lead to a gradual decrease in physical and mental capacity, a growing risk of disease, and ultimately death [68].

Thus, given that they are only somewhat connected to an individual’s age, these changes are neither linear nor consistent. Despite biological changes, ageing can frequently be attributed to other major life events like retirement, moving to a more suitable home, and the death of “significant others”, and South Africa, like many countries globally, is experiencing a significant demographic shift with the rapid growth of an ageing population [1,69]. Also, studies have shown that grandparents with higher education had lower odds of experiencing health challenges as they are more likely to have adequate and appropriate knowledge on how to prevent and manage these health conditions resulting from caregiving to their grandchildren, compared to their counterparts with no education [11,70]. This study’s findings in the unadjusted model showed consistency with past research that has found that grandparents with lower educational attainment may have poorer health than those with greater educational attainment [39,71]. This pattern is attributed to the large health inequalities brought about by education. However, the study findings in the adjusted model revealed that grandparents with higher education had higher odds of health challenges experienced as a result of being the primary caregiver to their grandchildren, compared to those with no or little education. This study finding is not consistent with previous studies, as few studies have indicated that educated grandparents experience health challenges owing to self-neglect. Recent studies have evidently stated that self-neglect is linked with adverse outcomes concerned with older adults’ physical [4,6] and psychological well-being [6,7], loss of dignity and self-esteem [7,38], illness [38,39], death [21,68] and healthcare utilisation [72,73]. 

Results from our study show that grandparents with low economic factors such as no regular salary or pension were related to increased chances of experiencing health challenges. This supports earlier studies, which posited that people with a lower socioeconomic status tend to be more prone to health issues that come from pressure and strenuous activities [1,5]. The reason grandparents with no regular salary or pension are plagued with health conditions when acting as the primary caregiver to their grandchildren may be associated with a “fear of the unknown” in trying to keep up with increased responsibilities associated with earning more. Studies have shown that many fears of grandparents without a regular salary or pension, who are caregivers to their grandchildren, can be traced to a negative experience that has been traumatic when proper care has not been given to their double orphaned grandchildren [7,35]. A few studies also showed that phobias can stem from a learned history, and many older adults are susceptible to being anxious about the unknown, which may lead to developing a fear of the unknown [23]. Moreover, this study found significant differences in the influence of health-related factors such as perceived health status and health consultation among grandparents in this study. For instance, the literature has shown that, over the years, poor perceived health status has been associated with increased odds of experiencing health challenges [19,22]. In agreement with these earlier findings, this study found that these grandparents with poor perceived health status were associated with higher odds of experiencing health challenges. In agreement with these earlier findings, this study found that perceived health status is associated with healthcare service utilisation and illness in developing countries [74,75]. Yet, little is known about the factors associated with perceived health status among grandparents who are primary caregivers to their double orphaned grandchildren [41,55].

Furthermore, grandparents who have never had a health consultation are more likely to experience health conditions. Studies have claimed that knowledge or information gained through interactions with individuals whose presence extends beyond the scope of a single medical visit may alter choices over time and affect behaviours [57,76]. According to different research, using alternate information sources may affect how well people communicate during consultations with healthcare providers [32,36]. For instance, in this era of the Internet and social media platforms, rich sources of information and expert knowledge can be made available to grandparents through Internet platforms if they have the facilities to access the Internet. Other studies have acknowledged a range of other people who can motivate positive and healthy communication with their “significant others” during healthcare consultations [4,11]. For example, a patient’s family, a doctor’s social and health network, and the media (radio, newspapers, and television) play an important role in grandparents’ clinic sessions. Thus, grandparents’ consultation on their health conditions is very important in influencing the improvement of their personal health with shared decision-making. 

### 4.1. Further Discussion: Insights from Changing Demography of Grandparenthood in South Africa

Demographic changes affect the time that individuals spend in different family roles, and one type of family relationship affected by early fertility is grandparenthood. Historically and in modern day societies, three-generation families are more common now than earlier, because children and grandchildren have higher chances of survival, and more people live long enough to see their grandchildren grow [77,78]. However, family formation patterns have also changed, as fertility declines, leading to increased childlessness; also, the postponement of marriage and childbearing affects the proportion of the population that ever-become grandparents, and the age at which grandparenthood begins for either younger or older age cohorts (See Appendix A (Table A1, Table A2, Table A3, Table A4, Table A5, Table A6, Table A7 and Table A8)). Thus, in contemporary South Africa, many families continue to undergo family transition and changes in family formation, with a range of challenges. A majority of South African families are being confronted with dual challenges of poverty and unemployment, making economic provision much more difficult in rural households. Since 1994, HIV/AIDS and TB, and more recently the COVID-19 pandemic [79,80], have placed families under significant strain, with the loss of caregivers and economic providers, but families in South Africa are characterised by significant resilience. 

However, being a grandparent relates to a life course, and is clearly defined by status, which determines and affects other stages in the life course, as being a grandparent is often linked to retirement. Yet, the transition to grandparenthood is associated with a change in status, roles, and identities which vary greatly in different contexts. However, the concepts of grandparenthood and ageing are related; the normative age at childbearing may be linked to the timing of grandparenthood and the social definition of ageing but may diverge from social expectations. Therefore, unlike ageing, grandparenting occurs “within a wider and more flexible age range” [48]. According to Statistics South Africa [45], more than 207 children were married, comprising 188 brides and 19 grooms, and these marriages were officially documented. Of the child marriages, 37 were registered as civil marriages and 19 were customary marriages [45].

In South Africa, younger adult grandparents aged 30–39 years (297 females and 40 males) have been documented in Statistics South Africa [48]. Many marriages conducted within the customs and traditions in rural communities were not documented by the Department of Home Affairs, leading to the under-reporting of cases of early child marriages in South Africa. Thus, several factors have been associated with the emergence of younger grandparents in South Africa such as increased child marriage [81], teenage pregnancies, *Ukuthwala* cultural practices, income generated from *lobola* negotiations, lack of accountability of community leaders towards child kidnapping, and religious beliefs. These factors have been shown to contribute to the demographic changes in the emergence of early grandparenthood in South Africa [48]. Regarding child marriage, Eastern and Southern Africa are among the regions with the highest prevalence of child marriage globally. At present, nearly one-third (32%) of the region’s young females were married before age 18 [82]. Concerning teenage pregnancies, Statistics South Africa [44] reported almost 34,000 teenage pregnancies, with 660 of those being girls under the age of 13 [83,84]. In South Africa, some of these teenage pregnancies have been linked to rape cases and arranged marriages [44]. 

Also, the prevalence of teenage pregnancies is high and is associated with rape and inappropriate sexual relationship among teens. Most of these teens do not have knowledge of the use of contraception or have access to sexual and reproductive health clinics. These barriers have led to an increased number of teenagers having children, resulting in their own children following the same path of becoming a teenage parent, and making their parents become young adult grandparents [44,84]. Also, *Ukuthwala* cultural practices have been reported to contribute to early grandparenthood, as it is a cultural form of abduction that involves kidnapping a girl or a young woman by a man and his friends or peers with the intention of compelling the woman’s family to endorse marriage negotiations [85]. Also, it was once an acceptable way for two young people in love to get married when their families opposed the match (and so was actually a form of elopement) [85,86]. Over time, *Ukuthwala* has been abused, however, “to victimize isolated rural women and enrich male relatives”, as older men are taking advantage of the cultural practices by marrying these children and sexually abusing them [86,87]. This type of cultural practice is common among the *Xhosa* and *Zulu* people from the Eastern Cape, Limpopo, and KwaZulu-Natal provinces [88]. 

Similarly, *lobola* payment is a cultural practice in South Africa where a bride price is paid to the bride’s family for her hand in marriage. These customs are sometimes abused and excused to erode human dignity and reinforce corrupt tendencies. This demeaning behaviour often handicaps the social welfare of a society, and *lobola* payment appears to be one of the most exploited praxes. Studies have shown that the identity of *lobola* has shifted from a token of appreciation to a commercial activity, where a family from a poor rural household generates income from the *lobola* negotiations without their female relative consenting to marriage [89,90]. Religious leaders’ frown upon children born outside wedlock, and pregnant teenagers are forced to enter into marriage, as illegitimacy is regarded as sin-related, with the stigma justified as a reprimand from God. This form of coercive behaviour has aided early child marriages due to pregnancy, without addressing the roots of early sexual initiation among teenagers. 

In South Africa, the rights of illegitimate children are protected and recognised by the Children’s Act of 2005. This law has abolished legal differences involving legitimate and illegitimate children, who are now treated equally in terms of inheritance rights [91,92]. Few studies have linked this intergenerational transition to the demography of grandparenthood. Demographic transitions of family formation are linked with the composition and transition of various family types. However, daughters of teenage mothers have been shown to be more likely to become teenage mothers at younger ages, linking teenage fertility to the family birth history [78]. According to the theory of socialisation, children born to teenage mothers have a higher chance of being teenage mothers, resulting in the inter-generational transmission of early childbearing, owing to factors such as reduced parenting, marital instability, and an environment of poor socio-economic conditions [93,94]. In South Africa, the fertility behaviour of teen mothers, such as their age at first birth, has been observed to influence the age at first birth of their daughters, as family disorganisation traits can be transmitted to their daughters by teen mothers [95]. 

### 4.2. Strengths and Limitations of the Study

This study has several major strengths and limitations. First, to the best of the authors’ knowledge, this is the first cross-sectional survey and nationally representative data that investigated the sociology and demography of ageing among grandparents who are caregivers to their double orphaned grandchildren. Second, the data analysis was basically conducted to determine the prevalence of grandparents who are caregivers to double orphans in South Africa, and associations based on the likelihood of the explanatory factors, but not providing a measure of causality; however, insight can be gained from using the 2017 National Income Dynamics Study (NIDS) wave 5 datasets from South Africa to improve the study’s generalisability to other settings or populations. Third, to the best of the authors’ knowledge, this is the first time that binary logistic regression models were aimed at elucidating the explanatory factors of the likelihood of the health outcomes of grandparents caring for double orphans in South Africa. There were some limitations, however, that need to be highlighted. First, owing to the nature of the study, we cannot draw causal inferences from the findings. This study also suggests the use of ethnographic methods that may unravel other possibilities that may influence the health outcomes of grandparents caring for their double orphaned grandchildren.

### 4.3. Implications for Social Gerontology and Demography of Ageing Research and Practice

The finding is consistent with previous studies that have found that grandparents with lower educational attainment may have poorer health than those with greater educational attainment. In most cases, grandparents have taken over the full responsibility of bringing up grandchildren as a result of unemployment, drug or alcohol abuse, or the death of the child’s parents. In South Africa, the aforementioned concern is exacerbated by changes in family structure owing to the severe impact of HIV/AIDS-related deaths, especially among young adult parents, leaving behind many orphaned children. This has brought about a change in roles for many grandparents, who have felt morally and culturally obliged to take care of their grandchildren, despite not being prepared for this parenting role. This study’s findings showed that there is a positive association between grandparents’ health outcomes and the role of caregiving to grandchildren, which agrees with several studies [4,10]. The growing number of grandparents as caregivers increases demands on the public health system and on medical and social services, due to adverse health conditions, which contribute to disability, diminished quality of life, and increased health- and long-term-care costs. Therefore, to address these social issues, insights from this study will be valuable to social and healthcare practitioners, who play a vital role in offering services to grandparents as caregivers to their grandchildren. There is a need for collaboration between various stakeholders and community health workers to empower and harness grandparents’ resilience to continue caring for their doubly orphaned grandchildren. Social gerontologists and demographers of ageing recognise the significance of collaboration and teamwork; therefore, their research and practices will go a long way to provide platforms for developing appropriate and adequate health interventions that will create welfare resources that will cater to the needs of grandparents taking the role of caregivers. Furthermore, policymakers, academics, and relevant role players will gain an in-depth understanding of this phenomenon from the South African context. 

## 5. Conclusions and Recommendations

Given the findings of this study, social gerontologists and demographers of ageing may identify the strengths and needs of grandparents as caregivers in order to determine the type of support system and services needed to improve their social and health welfare. This suggests that demographic, economic, and health-related factors are important in re-shaping health challenges experienced by grandparents as primary caregivers to double orphans in South Africa. Researchers and practitioners should incorporate these aspects in order to re-design strategic interventions and initiatives to develop future research that will tackle and address the health needs of these grandparents. Social gerontologists, demographers, and sociologists should collaborate to develop a platform of advocacy for the unique needs of grandparents providing care to their double orphaned grandchildren in order to improve the quality of the care by minimising the impact of age-related diseases and conditions, which vary depending on a person’s race, gender, and health. Furthermore, grandparents may be taught about their rights and responsibilities as well as the importance of sharing their social and health challenges with relevant community health workers, and family members whom they trust, to offload the burden of anxiety and worry. Lastly, policymakers should develop and implement policies that respond to the plight of grandparents caring for grandchildren who are doubly orphaned.

## Figures and Tables

**Table 1 ijerph-20-07158-t001:** Social demographic characteristics of grandparent caregivers stratified by sex of double orphans.

Characteristics	Grandparent Caregivers Caring for Male Double Orphans (*n* = 141,671)		Grandparent Caregivers for Female Double Orphans (*n* = 160,805)		Grandparent Caregivers for Both Sexes Double Orphans (*N* = 302,476)	
	Frequency	%	Frequency	%	Frequency	%
Demographics						
*Age group*						
25–34	34,921	24.6	42,784	26.6	77,706	25.7
35–44	25,624	18.1	39,765	24.7	65,390	21.6
45–54	25,915	18.3	31,931	19.9	57,847	19.1
55–64	29,006	20.5	22,485	14.0	51,491	17.0
65+	26,204	18.5	23,839	14.8	50,043	16.5
*Sex*						
Male	55,483	39.2	53,639	33.4	109,122	36.1
Female	86,188	60.8	107,166	66.6	193,354	63.9
*Population group*						
Black African	100,475	70.9	110,599	68.8	211,075	69.8
Non-Black African	41,195	29.1	50,206	31.2	91,401	30.2
*Education*						
No education	10,010	7.1	20,303	12.6	30,314	10.0
Primary	27,370	19.3	19,959	12.4	47,329	15.6
Secondary	59,300	41.9	82,422	51.3	141,723	46.9
Post-secondary	44,990	31.8	38,121	23.7	83,111	27.5
Economic-related						
*Regular salary*						
Yes	49,030	34.6	63,988	39.8	113,018	37.4
No	92,641	65.4	96,818	60.2	189,459	62.6
*Pension*						
Yes	28,807	20.3	29,389	18.3	58,196	19.2
No	112,863	79.7	131,416	81.7	244,280	80.8
Health-related						
*Depression in the past week*						
No	80,914	57.1	103,499	64.4	184,413	61.0
Yes	60,757	42.9	57,307	35.6	118,063	39.0
*Perceived health status*						
Excellent	61,732	43.6	88,225	54.9	149,957	49.6
Good	48,681	34.4	43,628	27.1	92,309	30.5
Poor	31,257	22.1	28,952	18.0	60,210	19.9
*Last health consultation*						
Never	2799	2.0	12,061	7.5	14,860	4.9
In the last month	92,740	65.5	99,319	61.8	192,059	63.5
Last year and more	46,131	32.6	49,425	30.7	95,557	31.6
*Medical aid*						
Yes	27,203	19.2	30,304	18.8	57,507	19.0
No	114,468	80.8	130,501	81.2	244,969	81.0
Geographical type						
*Geographical area*						
Rural	40,496	28.6	59,856	37.2	100,352	33.2
Urban	101,175	71.4	100,950	62.8	202,124	66.8
*Province*						
Western Cape	12,158	8.6	32,574	20.3	44,732	14.8
Eastern Cape	13,806	9.7	20,264	12.6	34,070	11.3
Northern Cape	3152	2.2	2324	1.4	5475	1.8
Free State	5040	3.6	6433	4.0	11,472	3.8
KwaZulu-Natal	21,152	14.9	32,249	20.1	53,401	17.7
North West	7624	5.4	6437	4.0	14,061	4.6
Gauteng	58,042	41.0	34,757	21.6	92,799	30.7
Mpumalanga	12,868	9.1	6657	4.1	19,525	6.5
Limpopo	7829	5.5	19,111	11.9	26,940	8.9

Source: Authors’ Compilation, 2023.

**Table 2 ijerph-20-07158-t002:** Reported health challenges by grandparents as caregivers to double orphans.

Reported Health Challenges	No		Yes	
	Frequency	%	Frequency	%
Painful urination	298,781	98.8	3695	1.2
Severe weight loss	295,357	97.7	7119	2.4
Diarrhoea	281,471	93.1	21,005	6.9
Chest pain	279,806	92.5	22,670	7.5
Swelling of ankles	274,165	90.6	28,311	9.4
Cough	273,595	90.5	28,881	9.6
Joint pain/arthritis	243,178	80.4	59,298	19.6
Backache	242,262	80.1	60,214	19.9
Body ache	241,566	79.9	60,775	20.1
Fever	222,742	73.6	79,734	26.4
Headache	220,981	73.1	81,495	26.9

Source: Authors’ Compilation, 2023.

**Table 3 ijerph-20-07158-t003:** Bivariate results of grandparent caregivers of double orphans and their associated socio-demographic factors by health challenges experienced and stratified by orphan gender.

Factors	Caring for Males						Caring for Females						Caring for Both Sexes					
	No		Yes				No		Yes				No		Yes			
Demographics	Freq.	%	Freq.	%	χ^2^	p	Freq.	%	Freq.	%	χ^2^	p	Freq.	%	Freq.	%	χ^2^	p
*Age group*					15.63	0.00 *					22.76	0.00 *					33.97	0.00 *
25–34	17,433	49.9	17,488	50.1			25,240	59.0	17,545	41.0			42,673	54.9	35,033	45.1		
35–44	14,144	55.2	11,480	44.8			27,729	69.7	12,037	30.3			41,873	64.0	23,517	36.0		
45–54	10,973	42.3	14,943	57.7			9788	30.7	22,143	69.3			20,761	35.9	37,086	64.1		
55–64	6103	21.0	22,903	79.0			4407	19.6	18,078	80.4			10,510	20.4	40,981	79.6		
65+	1203	4.6	25,001	95.4			4957	20.8	18,882	79.2			6160	12.3	43,883	87.7		
*Sex*					0.04	0.84					2.38	0.12					1.35	0.25
Male	15,594	28.1	39,889	71.9			29,626	55.2	24,013	44.8			45,220	41.4	63,902	58.6		
Female	34,261	39.8	51,927	60.2			42,495	39.7	64,671	60.3			76,756	39.7	116,598	60.3		
*Population group*					0.71	0.40					0.00	0.99					0.32	0.57
Black African	37,657	37.5	62,819	62.5			51,659	46.7	58,940	53.3			89,316	42.3	121,759	57.7		
Non-Black African	12,199	29.6	28,997	70.4			20,462	40.8	29,744	59.2			32,661	35.7	58,741	64.3		
*Highest education*					6.91	0.08					9.62	0.02 *					15.17	0.00 *
No education	3490	34.9	6521	65.1			5528	27.2	14,776	72.8			9017	29.7	21,296	70.3		
Primary	4610	16.8	22,760	83.2			6575	32.9	13,384	67.1			11,185	23.6	36,144	76.4		
Secondary	25,587	43.1	33,713	56.9			38,674	46.9	43,749	53.1			64,261	45.3	77,462	54.7		
Post-secondary	16,169	35.9	28,822	64.1			21,345	56.0	16,777	44.0			37,513	45.1	45,598	54.9		
**Economic**																		
*Regular salary*					5.74	0.02 *					3.19	0.07					8.31	0.00 *
Yes	25,697	52.4	23,333	47.6			31,163	48.7	32,825	51.3			56,860	50.3	56,158	49.7		
No	24,159	26.1	68,482	73.9			40,958	42.3	55,860	57.7			65,117	34.4	124,342	65.6		
*Pension*					5.82	0.02 *					4.25	0.04 *					9.62	0.00 *
Yes	1851	6.4	26,957	93.6			8144	27.7	21,245	72.3			9995	17.2	48,202	82.8		
No	48,005	42.5	64,859	57.5			63,977	48.7	67,439	51.3			111,982	45.8	132,298	54.2		
Health-related																		
*Depression in the past week*					0.91	0.34					2.50	0.11					3.38	0.07
No	21,421	26.5	59,493	73.5			51,934	50.2	51,565	49.8			73,355	39.8	111,058	60.2		
Yes	28,434	46.8	32,323	53.2			20,187	35.2	37,119	64.8			48,621	41.2	69,442	58.8		
*Perceived health status*					22.44	0.00 *					21.98	0.00 *					43.75	0.00 *
Excellent	35,550	57.6	26,182	42.4			54,583	61.9	33,642	38.1			90,134	60.1	59,824	39.9		
Good	9078	18.6	39,603	81.4			14,244	32.6	29,384	67.4			23,322	25.3	68,987	74.7		
Poor	5227	16.7	26,031	83.3			3294	11.4	25,659	88.6			8521	14.2	51,689	85.8		
*Last health consultation*					15.06	0.00 *					14.56	0.00 *					28.64	0.00 *
Never	2799	21.0	10,520	79.0			1541	0.0	0.0	0.0			4340	29.2	10,520	70.8		
In the last month	28,374	30.6	64,366	69.4			35,006	35.2	64,313	64.8			63,381	33.0	128,679	67.0		
Last year and more	18,682	40.5	27,450	59.5			26,595	53.8	22,831	46.2			45,276	47.4	50,280	52.6		
*Medical aid*					0.00	0.97					0.00	0.98					0.00	0.96
Yes	10,472	38.5	16,731	61.5			11,342	37.4	18,962	62.6			21,814	37.9	35,694	62.1		
No	39,384	34.4	75,084	65.6			60,780	46.6	69,722	53.4			100,163	40.9	144,806	59.1		
**Geographical type**																		
*Geographical area*					0.07	0.79					0.02	0.89					0.08	0.78
Rural	19,161	47.3	21,335	52.7			26,444	44.2	33,412	55.8			45,605	45.4	54,747	54.6		
Urban	30,694	30.3	70,481	69.7			45,677	45.2	55,272	54.8			76,372	37.8	125,753	62.2		
*Province*					6.58	0.58					7.58	0.48					5.96	0.65
Western Cape	3101	25.5	9057	74.5			7975	24.5	24,599	75.5			11,076	24.8	33,656	75.2		
Eastern Cape	5219	37.8	8588	62.2			12,091	59.7	8173	40.3			17,309	50.8	16,761	49.2		
Northern Cape	917	29.1	2234	70.9			816	35.1	1508	64.9			1733	31.7	3742	68.3		
Free State	1267	25.1	3773	74.9			1582	0.0	4851	0.0			2849	24.8	8623	75.2		
KwaZulu-Natal	6858	32.4	14,294	67.6			17,805	55.2	14,444	44.8			24,662	46.2	28,738	53.8		
North West	1043	13.7	6581	86.3			1186	18.4	5251	81.6			2229	15.9	11,832	84.1		
Gauteng	22,288	38.4	35,755	61.6			21,412	61.6	13,346	38.4			43,699	47.1	49,100	52.9		
Mpumalanga	3630	28.2	9237	71.8			1258	18.9	5399	81.1			4888	25.0	14,637	75.0		
Limpopo	5533	70.7	2297	29.3			7997	41.8	11,114	58.2			13,529	50.2	13,411	49.8		

Source: Authors’ Compilation, 2023. Abbreviations: Freq. = frequency; % = percentage; χ^2^= chi-square; **p** = *p*-value; * (asterisk) = significant.

**Table 4 ijerph-20-07158-t004:** Multilevel logistic analysis of unadjusted predictors of health challenges experienced as caregivers to double orphans.

Health Challenges	Male		Female		Both Sexes	
Demographics	UOR (95% CI)	p	UOR (95% CI)	p	UOR (95% CI)	p
*Age group*						
25–34 (RC)						
35–44	1.04 (0.46–2.36)	0.93	0.82 (0.34–1.96)	0.66	0.96 (0.53–1.73)	0.89
45–54	1.81 (0.73–4.51)	0.20	2.20 (0.90–5.42)	0.09	* 2.01 (1.06–3.82)	0.03
55–64	2.42 (0.92–6.37)	0.08	* 7.51 (2.52–22.44)	0.00	* 4.16 (2.04–8.48)	0.00
65+	* 8.70 (2.31–32.78)	0.00	* 3.25 (1.26–8.37)	0.02	* 4.65 (2.20–9.80)	0.00
*Sex*						
Male (RC)						
Female	1.06 (0.58–1.95)	0.84	1.62 (0.88–3.01)	0.12	1.29 (0.84–1.98)	0.25
*Population group*						
Black African (RC)						
Non-Black African	0.74 (0.37–1.49)	0.40	1.01 (0.49–2.05)	0.99	0.87 (0.53–1.43)	0.57
*Highest education*						
No education (RC)						
Primary	1.20 (0.33–4.42)	0.78	3.12 (0.98–10.01)	0.06	2.09 (0.90–4.86)	0.09
Secondary	0.53 (0.17–1.65)	0.28	0.79 (0.33–1.88)	0.59	0.70 (0.36–1.39)	0.31
Post-secondary	0.38 (0.12–1.23)	0.11	0.68 (0.26–1.78)	0.44	0.56 (0.27–1.15)	0.12
*Economic*						
*Regular salary*						
Yes (RC)						
No	* 2.12 (1.14–3.92)	0.02	1.78 (0.08–0.94)	3.35	* 1.90 (1.23–2.95	0.00
*Pension*						
Yes (RC)						
No	* 0.32 (0.12–0.84)	0.02	* 0.45 (0.21–0.97)	0.04	* 0.40 (0.22–0.72)	0.00
Health-related						
Depression in the past week						
No (RC)						
Yes	1.34 (0.73–2.44)	0.34	1.62 (0.89–2.96)	0.12	1.49 (0.97–2.27)	0.07
*Perceived health status*						
Excellent (RC)						
Good	* 4.33 (1.99–9.42)	0.00	* 3.56 (1.74–7.27)	0.00	* 3.85 (2.28–6.49)	0.00
Poor	* 5.87 (2.07–16.61)	0.00	* 5.79 (2.29–14.62)	0.00	* 5.73 (2.88–11.43)	0.00
*Last health consultation*						
Never (RC)						
In the last month	* 2.15 (1.14–4.04)	0.02	8.77 (1.81–42.58)	0.01	* 15.74 (3.50–70.70)	0.00
Last year and more	1.00	-	3.52 (0.70–17.75)	0.13	* 6.92 (1.52–31.55)	0.01
*Medical aid*						
Yes (RC)						
No	0.99 (0.46–2.14)	0.97	1.01 (0.43–2.37)	0.98	0.98 (0.56–1.74)	0.96
Geographical type						
*Geographical area*						
Rural (RC)						
Urban	1.09 (0.56–2.12)	0.79	1.05 (0.54–2.01)	0.89	1.07 (0.67–1.70)	0.78
*Province*						
Western Cape (RC)						
Eastern Cape	2.40 (0.52–10.99)	0.26	0.45 (0.13–1.50)	0.19	0.85 (0.34–2.14)	0.73
Northern Cape	6.00 (0.93–38.63)	0.06	1.50 (0.25–8.98)	0.66	2.63 (0.74–9.33)	0.14
Free State	2.00 (0.32–12.33)	0.46	0.83 (0.16–4.30)	0.83	1.13 (0.34–3.74)	0.85
KwaZulu-Natal	2.25 (0.67–7.56)	0.19	0.53 (0.19–1.46)	0.22	0.94 (0.44–2.01)	0.88
North West	4.50 (0.85–23.80)	0.08	1.50 (0.25–8.98)	0.66	2.25 (0.69–7.34)	0.18
Gauteng	2.31 (0.66–8.03)	0.19	0.46 (0.15–1.44)	0.18	0.93 (0.42–2.08)	0.86
Mpumalanga	1.29 (0.29–5.77)	0.74	2.00 (0.35–11.44)	0.44	1.17 (0.41–3.29)	0.77
Limpopo	1.50 (0.27–8.45)	0.65	0.90 (0.23–3.49)	0.88	1.08 (0.38–3.09)	0.88

Source: Authors’ Compilation, 2023. Abbreviations: 95% Conf. Int = 95% confidence interval; UOR = unadjusted odds ratio; **p** = *p*-value; * (asterisk) = significant.

**Table 5 ijerph-20-07158-t005:** Multilevel logistic analysis of adjusted predictors of health challenges experienced as caregivers to double orphans.

Health Challenges	Male		Female		Both Sexes	
Demographics	AOR (95% CI)	p	AOR (95% CI)	p	AOR (95% CI)	p
*Age group*						
25–34 (RC)						
35–44	0.73 (0.22–2.36)	0.60	1.33 (0.39–4.47)	0.65	1.03 (0.48–2.24)	0.93
45–54	1.94 (0.47–8.09)	0.36	1.59 (0.45–5.71)	0.47	1.67 (0.72–3.86)	0.23
55–64	2.50 (0.43–14.56)	0.31	* 12.04 (1.55–93.25)	0.02	* 5.04 (1.54–16.50)	0.01
65+	* 27.34 (2.19–341.92)	0.01	6.73 (0.53–85.19)	0.14	* 8.86 (1.90–41.28)	0.01
*Sex*						
Male (RC)						
Female	* 0.34 (0.13–0.89)	0.03	1.44 (0.59–3.51)	0.42	0.85 (0.48–1.52)	0.59
*Population group*						
Black African (RC)						
Non-Black African	* 0.22 (0.06–0.79)	0.02	0.90 (0.32–2.55)	0.85	0.61 (0.30–1.25)	0.18
*Education*						
No education (RC)						
Primary	6.44 (0.85–48.66)	0.07	* 13.94 (2.21–87.83)	0.01	* 4.79 (1.49–15.39)	0.01
Secondary	1.84 (0.35–9.77)	0.48	3.03 (0.69–13.40)	0.14	1.79 (0.67–4.78)	0.25
Post-secondary	1.09 (0.17–6.81)	0.93	3.13 (0.65–15.09)	0.16	1.44 (0.49–4.21)	0.51
*Regular salary*						
Yes (RC)						
No	* 4.68 (1.59–13.84)	0.01	1.70 (0.60–4.84)	0.32	* 2.01 (1.06–3.83)	0.03
Economic						
*Pension*						
Yes (RC)						
No	5.05 (0.68–37.74)	0.11	3.93 (0.47–32.81)	0.21	* 3.95 (1.05–14.89)	0.04
*Depression in the past week*						
No (RC)						
Yes	0.80 (0.32–1.98)	0.63	1.56 (0.63–3.88)	0.33	1.03 (0.58–1.83)	0.93
Health-related						
*Perceived health status*						
Excellent (RC)						
Good	* 6.15 (1.98–19.14)	0.00	2.23 (0.86–5.81)	0.10	* 2.92 (1.49–5.73)	0.00
Poor	* 4.28 (1.00–18.35)	0.05	* 4.00 (0.99–16.18)	0.05	* 3.30 (1.35–8.09)	0.01
*Last health consultation*						
Never (RC)						
In the last month	2.03 (0.80–5.13)	0.14	6.60 (0.57–76.42)	0.13	* 12.87 (1.44–115.31)	0.02
Last year and more	-	-	2.81 (0.22–35.39)	0.43	6.23 (0.69–55.78)	0.10
*Medical aid*						
Yes (RC)						
No	0.22 (0.04–1.10)	0.07	1.32 (0.36–4.88)	0.68	0.86 (0.35–2.09)	0.74
Geographical type						
*Geographical area*						
Rural (RC)						
Urban	1.63 (0.55–4.79)	0.38	1.25 (0.43–3.61)	0.69	1.36 (0.69–2.67)	0.38
Western Cape (RC)						
Eastern Cape	1.12 (0.16–7.79)	0.91	0.46 (0.10–2.22)	0.34	0.87 (0.28–2.66)	0.80
Northern Cape	3.45 (0.35–34.37)	0.29	0.65 (0.07–6.12)	0.71	1.98 (0.44–8.83)	0.37
Free State	1.74 (0.19–16.35)	0.63	0.54 (0.07–4.45)	0.57	0.92 (0.23–3.68)	0.90
KwaZulu-Natal	1.75 (0.36–8.52)	0.49	0.32 (0.07–1.38)	0.13	0.97 (0.38–2.49)	0.95
North West	4.97 (0.53–46.83)	0.16	1.33 (0.14–12.96)	0.81	2.72 (0.63–11.79)	0.18
Gauteng	1.29 (0.27–6.23)	0.75	0.43 (0.10–1.82)	0.25	0.87 (0.34–2.22)	0.78
Mpumalanga	1.30 (0.19–8.84)	0.79	1.08 (0.14–8.14)	0.94	1.26 (0.38–4.19)	0.70
Limpopo	0.40 (0.03–4.75)	0.47	0.93 (0.16–5.56)	0.94	0.99 (0.26–3.76)	0.99

Source: Authors’ Compilation, 2023. Abbreviations: 95% Conf. Int = 95% confidence interval; AOR = adjusted odds ratio; **p** = *p*-value; * (asterisk) = significant.

## Data Availability

Data are from the Demographic and Health Survey and the dataset is open to qualified researchers free of charge. In order to access the data from NIDS-CRAM survey, a written request was submitted to the NID-CRAM and permission was granted to use the data for this survey. To request access to the dataset, please apply at http://www.nids.uct.ac.za/nids-data/documentation/overview-documentation/wave-5.

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
