# Peer review of "“Health Outcomes of Grandparents Caring for Double Orphans in South Africa”: What Are the Determinants?"

_ijerph, 2023, doi:10.3390/ijerph20247158_

Round 1
Reviewer 1 Report
Comments and Suggestions for Authors
Through detailed investigation and research, the paper reveals a phenomenon that has become a reality, that is, globally, more and more grandparents are taking on the responsibility of raising grandchildren. Therefore, the research in this paper has strong sociological significance, not only for South Africa, but also for all governments around the world to formulate policies related to social welfare, elderly care, and childbirth. The research method of the paper is appropriate, the data is detailed, and the conclusions are clear. It is recommended to accept the paper for publication. To make the paper more readable, here are some suggestions:
1. In the second part, it is recommended to move the introduction about the geographical and sociodemographic background of South Africa to the first part.
2. It is recommended that the author further introduce the NIDS dataset.
3. In the fourth part, it seems that there should not be too many references to previous research results, and here should mainly state some discussions based on the research paper. If it is necessary to discuss the previous research, it is recommended to include it in the first part of the paper.
Author Response
Review Report (Reviewer 1)
Comments and Suggestions for Authors
Through detailed investigation and research, the paper reveals a phenomenon that has become a reality, that is, globally, more and more grandparents are taking on the responsibility of raising grandchildren. Therefore, the research in this paper has strong sociological significance, not only for South Africa, but also for all governments around the world to formulate policies related to social welfare, elderly care, and childbirth. The research method of the paper is appropriate, the data is detailed, and the conclusions are clear. It is recommended to accept the paper for publication. To make the paper more readable, here are some suggestions:
- In the second part, it is recommended to move the introduction about the geographical and sociodemographic background of South Africa to the first part.
Authors’ response: Thank you very much for your comments. The aspect that discusses demographics of South African in the introduction section was to clearly show the relationship between the social issues associated with the demographic factor, that accounts for high level of grandparenting as caregivers in South Africa [See Page 3 in Red colour ink].
While the section that talks about the geographical and sociodemographic background of South Africa under the method section shows the background context of South Africa and how it has played a role in in the development of grandparenting phenomenon, quite different from other African countries [See Page 4 to Page 5 in Red colour ink].
- It is recommended that the author further introduce the NIDS dataset.
Author’s response: Thank you very much for your comments. The NIDS dataset was introduced under the section of study design and data source [See Page 5 in Red colour ink].
- In the fourth part, it seems that there should not be too many references to previous research results, and here should mainly state some discussions based on the research paper. If it is necessary to discuss the previous research, it is recommended to include it in the first part of the paper.
Authors’ response: Thank you very much for your comments. The many references was as a result of wide-ranging research on the phenomenon of this study. The plenty of references in our research showed that the authors have done extensive research on the topic and that we have a good understanding of the subject matter. It has help us to support our arguments and ideas with evidence from other sources, which can make our work more persuasive and convincing. This will help us to avoid plagiarism by ensuring that we acknowledge the work of others and provide a reference to the source of the information we used in this study. The references we used are relevant, reliable, and up-to-date. We ensured that we use a variety of sources, including books, journal articles, and online resources, to provide a well-rounded and comprehensive view of the topic. Also, we discuss the previous research in line with the references used in this study and all the references listed were used in this study [See Page 27 to Page 31 in Red colour ink].

Reviewer 2 Report
Comments and Suggestions for Authors
Abstract:
Keywords in accordance with MESH terms (https://www.ncbi.nlm.nih.gov/mesh/):
1. Please provide keywords in accordance with MESH terms.
Methods and Materials:
1. For the sections "Study Design and Data Source" and "Study Population and Sample Size," it would be beneficial to include a flowchart depicting the 5 waves of the National Income Dynamics Study (NIDS), illustrating the number of participants in each wave and demonstrating the selection process for the study population.
Results:
1. In Figure 1, Figure 2, and Figure 3, why does the y-axis show prevalence (%) of less than 100%?
2. What is the significance of Figure 1, Figure 2, and Figure 3 if all the data are described in the text? These figures appear to be redundant and lack informativeness.
3. You mention "Graphs" in the "Statistical Analysis" section, but there are no corresponding figures labeled as Graph 1, Graph 2, or Graph 3 in the article.
4. In Table 2, some numbers do not match the specified frequencies.
5. Is there a statistically significant difference between groups in Figure 1 and Figure 2? If so, it would be appropriate to indicate p-values on the graphs.
6. I recommend aligning the presentation of Odd ratio and their confidence intervals in Table 4 with the format used in the article provided at https://link.springer.com/article/10.1186/s12887-020-02467-1.
7. What was the purpose of including all the variables from the bivariate regression in the multivariate regression analysis?
8. In my opinion, some sections of the introduction, results (e.g., "Prevalence of grandparents as caregivers to double orphans by sex," "Prevalence of grandparents as caregivers to double orphans by age," etc.), and the discussion should be moved to supplementary files due to the article's information overload.
Author Response
Review Report (Reviewer 2)
Comments and Suggestions for Authors
Abstract:
- Keywords in accordance with MESH terms (https://www.ncbi.nlm.nih.gov/ mesh/): Please provide keywords in accordance with MESH terms.
Authors’ response: Thank you very much for your comments. We have addressed the keywords in accordance with MESH terms from the online site (https://www.nc bi.nlm.nih.gov/mesh/?term=). Then the revised Keywords include Aging, demography, caregivers, grandparents, gerontology, orphans [See Page 1 in Blue colour ink].
Methods and Materials:
- For the sections "Study Design and Data Source" and "Study Population and Sample Size," it would be beneficial to include a flowchart depicting the 5 waves of the National Income Dynamics Study (NIDS), illustrating the number of participants in each wave and demonstrating the selection process for the study population.
Authors’ response: Thank you very much for your comments. A flowchart depicting the 5 Waves of the National Income Dynamics Study (NIDS), illustrating the number of participants in each wave and demonstrating the selection process for the study population has been inserted in the main manuscript [See Page 6 in Blue colour ink].
Results:
- In Figure 1, Figure 2, and Figure 3, why does the y-axis show prevalence (%) of less than 100%?
Authors’ response: Thank you very much for your comments. We are aware that the percentage must be expressed as 100. However, the figure we have is 20%, and if we expressed the graph in 100% scale, the position of the 20% will be very small, and or it will not be very clear to the reader [See Page 10 to Page 11 in Blue colour ink].
- What is the significance of Figure 1, Figure 2, and Figure 3 if all the data are described in the text? These figures appear to be redundant and lack informativeness.
Authors’ response: Thank you very much for your comments. We have retained the text in the main manuscript and we have address the issues associated with graphs (Fig. 4, Fig 5 and Fig 6) and we have added 3 more graphs (Fig. 1, Fig. 2, and Fig. 3). Since all the data from the graph were described in the text, we have moved all the graphs to the Appendix 2 [See Page 9 to Page 10; and See Page 41 to Page 43 for Appendix 1 and Appendix 2 in Blue colour ink].
- You mention "Graphs" in the "Statistical Analysis" section, but there are no corresponding figures labeled as Graph 1, Graph 2, or Graph 3 in the article.
Authors’ response: Thank you very much for your comments. This has been addressed and Graphs are replaced with Figures [See Page 41 to Page 43 in Blue colour ink].
- In Table 2, some numbers do not match the specified frequencies.
Authors’ response: Thank you very much for your comments. This has been addressed and the numbers have been addressed to match the specified frequencies [See Page 10 in Blue colour ink].
- Is there a statistically significant difference between groups in Figure 1 and Figure 2? If so, it would be appropriate to indicate p-values on the graphs.
Authors’ response: Thank you very much for your comments. All the Figures (2-7) were done in descriptive, and the aim was not to test it at that level in order to evaluate a statistically significant difference between groups in Figure 5 and 6. The aim was done in 3 levels – descriptive, bivariate and inferential. In this aspect of the Figures, we did not engage in inferential analysis regarding the Figures or graphs of this study [See Page 41 to Page 43 in Blue colour ink].
- I recommend aligning the presentation of Odd ratio and their confidence intervals in Table 4 with the format used in the article provided at https://link.springer.com /article/10.1186/s12887-020-02467-1.
Authors’ response: Thank you very much for your comments. Table 4 and Table 5 have been aligned with the presentation of Odd ratio and their confidence intervals with the format used in the articles you provided the link: https://link.springer.com /article/10.1186/s12887-020-02467-1 [See Table 4 and Table 5 in Page 16 to Page 19 in Blue colour ink].
- What was the purpose of including all the variables from the bivariate regression in the multivariate regression analysis?
Authors’ response: Thank you very much for your comments. The variables that were not significant at bivariate analysis were also considered in the multivariate analysis because the interactions at the multivariate level will all have equal chances to express themselves as a predictor. Sometimes those variables who are not significant at bivariate levels will become significant at the multivariate levels, as through interactions of other variables will make or help to be become strong and become significant in the multivariate analyses. Thus, some of the variables at the bivariate level that were not significant, through interactions of other variables , it becomes significant. [See Table 3 in Page 11 to Page 14 in Blue colour ink].
- In my opinion, some sections of the introduction, results (e.g., "Prevalence of grandparents as caregivers to double orphans by sex," "Prevalence of grandparents as caregivers to double orphans by age," etc.), and the discussion should be moved to supplementary files due to the article's information overload.
Authors’ response: Thank you very much for your comments. All the Tables have been moved to the Appendix 1 and all the figures have been move to the Appendix 2. [See Page 31 to Page 42 in Blue colour ink].

Reviewer 3 Report
Comments and Suggestions for Authors
Dear authors, first of all, I must congratulate you on the topic addressed. Second, this article is well constructed in its bulk; the only doubt remains with the concept of Determinants that appears in the title and objective but is not developed in results or discussions.
Title: highlights the lack of the concept of Determinants in the results and discussions, although it appears objective and methodology; determinants are not detailed in the results or the arguments.
Summary: The objective in the summary must be the same as stated throughout the text. The date of data collection and sample eligibility criteria must be declared.
MeSH: you must check that the keywords are MeSH.
Results: the tables must detail the source, and if made by the authors, they must also indicate it.
Discussion: it is requested that the countries where the research is carried out be listed in such a way as to understand the level of comparison of the results.
No
Author Response
Review Report (Reviewer 3)
Comments and Suggestions for Authors
Dear authors, first of all, I must congratulate you on the topic addressed. Second, this article is well constructed in its bulk; the only doubt remains with the concept of Determinants that appears in the title and objective but is not developed in results or discussions.
Title: highlights the lack of the concept of Determinants in the results and discussions, although it appears objective and methodology; determinants are not detailed in the results or the arguments.
Authors’ response: Thank you very much for your comments. In Demography and Sociology fields, determinants are factors that influence the outcome of interest in a study. Demographers and Sociologists use these determinants (or factors) to study population characteristics such as age, sex, race, and other demographic variables. (See this chapter for reference: Baker J, Swanson DA, Tayman J, Tedrow LM (2017). Basic demographic concepts. In: Cohort change ratios and their applications. Springer, Cham. https://doi.org/10.100/978-3-319-53745-0_2.
In this study determinants (factors) were discussed in the variable measurement (independent variables in the Method section (Page 7) and in the Discussion section (Page 20-Page 24). The determinants discussed in this study were classified into demographics (age, sex, population group, highest education), Economic (regular salary, pension), Health-related (depression in the past week, perceived health status, last health consultation, medical aid) and geographical type (geographical area, province). See all the aforementioned determinants analysed in Table 1 (See Page 8 to Page 9), Table 3 (See Page 12 to Page 14), Table 4 (See Page 16 to Page 17), and Table 5 (See Page 18 to Page 19).
Summary: The objective in the summary must be the same as stated throughout the text. The date of data collection and sample eligibility criteria must be declared.
Authors’ response: Thank you very much for your comments. The date of the data was only mentioned in year such as 2017 NIDS (See Page 5). The objective in the summary is the same as stated throughput the text and the date of the data collection and sample eligibility criteria (inclusion and exclusion criteria) was inserted in the Methods. (See Page 7 to Page 8).
MeSH: you must check that the keywords are MeSH.
Authors’ response: Thank you very much for your comments. We have addressed the keywords in accordance with MESH terms from the online site (https://www.nc bi.nlm.nih.gov/mesh/?term=). Then the revised Keywords include Aging, demography, caregivers, grandparents, gerontology, orphans [See Page 1 in Blue colour ink].
Results: the tables must detail the source, and if made by the authors, they must also indicate it.
Authors’ response: Thank you very much for your comments. The tables (Table 1 to 5) were made by the Authors and we have inserted the source as: “Source: Authors’ Compilation” (See Table 1 in Page 8 to Page 9 in Brown colour ink), (See Table 3 in Page 12 to Page 14 in Brown colour ink), (See Table 4 in Page 16 to Page 17 in Brown colour ink), and (See Table 5 in Page 18 to Page 19 in Brown colour ink).
Discussion: it is requested that the countries where the research is carried out be listed in such a way as to understand the level of comparison of the results.
Authors’ response: Thank you very much for your comments. The countries listed or cited to illustrate our findings and to show that the level of comparison of the results are shown in counties where grandparents as caregivers are culturally or traditionally practiced. This phenomenon of grandparents as caregivers is being seen practiced in developed and developing countries.

Round 2
Reviewer 2 Report
Comments and Suggestions for Authors
You made changes based on some of the comments in the previous review. However, some comments remain unchanged. I kindly request your attention to those comments
1. You have corrected the keywords according to MESH terms, but there is some discrepancy. Therefore, I kindly request you to review them once again and make the necessary adjustments.
2. In Figure 2, Figure 3, and Figure in appendix, in y-axis should show (%) from 0 to 100%?
3. Is there a statistically significant difference between groups in Figure 3-7 in appendix? If so, it would be appropriate to indicate p-values on the graphs.
Author Response
Peer Reviewer 2 - Comments and Suggestions for Authors
You made changes based on some of the comments in the previous review. However, some comments remain unchanged. I kindly request your attention to those comments.
- You have corrected the keywords according to MESH terms, but there is some discrepancy. Therefore, I kindly request you to review them once again and make the necessary adjustments.
Authors’ response: Thank you very much for your comments. We have corrected the keywords according to the MESH terms as the Peer Reviewer 2 had previously suggested. The revised keywords we got according to our title and abstract was included in our previous submission: Aging, demography, caregivers, grandparents, gerontology, orphans [See the corrections in Blue colour ink in Page 1].
Therefore, kindly indicate in detail the discrepancy you observe and how to go about it.
- In Figure 2, Figure 3, and Figure in appendix, in y-axis should show (%) from 0 to 100%?
Authors’ response: Thank you very much for your comments. We explain in our previous response that there is no graph that has a scale of 100% when the variables you are measuring is not up to 100%. Yes, we know that percentages are expressed as 100%, which we have done in all the graphs, however, when drawing the graph in Excel sheet, the graph has its scale and it will scale according to the class boundaries – Lower class boundary and Upper-class boundary.
- Is there a statistically significant difference between groups in Figure 3-7 in appendix? If so, it would be appropriate to indicate p-values on the graphs.
Authors’ response: Thank you very much for your comments. We explain in our previous response that all the figures 3 to Figure 7 were all done at the level of descriptive analysis. The objective of having the graphs in this study was not to establish a statistically significant difference between groups (Figure 3 to Figure 7). This was indicated in the analysis section of the methods of this study [See Page 41 to Page 43 in Blue colour ink].
Therefore, we do not know any other way of drawing a descriptive graph to show statistically significance when already have Table 2 – the bivariate analysis for statistically significance of the outcome variable and the explanatory variables.
